# Preparation of N-Doped Layered Porous Carbon and Its Capacitive Deionization Performance

**DOI:** 10.3390/ma16041435

**Published:** 2023-02-08

**Authors:** Rui Liu, Shouguang Yao, Yan Shen, Yu Tian, Qiqi Zhang

**Affiliations:** School of Energy and Power Engineering, Jiangsu University of Science and Technology, Zhenjiang 212000, China

**Keywords:** capacitive deionization, N-doped layered porous carbon, salt adsorption capacity

## Abstract

In this study, N-doped layered porous carbon prepared by the high-temperature solid-state method is used as electrode material. Nano calcium carbonate (CaCO_3_) (40 nm diameter) is used as the hard template, sucrose (C_12_H_22_O_11_) as the carbon source, and melamine (C_3_H_6_N_6_) as the nitrogen source. The materials prepared at 850 °C, 750 °C, and 650 °C are compared with YP-50F commercial super-activated carbon from Japan Kuraray Company. The electrode material at 850 °C pyrolysis temperature has a higher specific surface area and more pores suitable for ion adsorption. Due to these advantages, the salt adsorption capacity (SAC) of the N-doped layered porous carbon at 850 °C reached 12.56 mg/g at 1.2 V applied DC voltage, 500 mg/L initial solution concentration, and 15 mL/min inlet solution flow rate, which is better than the commercial super activated carbon as a comparison. In addition, it will be demonstrated that the N-doped layered porous carbon at 850 °C has a high salt adsorption capacity CDI performance than YP-50F by studying parameters with different applied voltages and flow rates as well as solution concentrations.

## 1. Introduction

With the acceleration of industrialization, the pollution of water resources is becoming more and more serious. There is a scarcity of freshwater resources on Earth that can be directly used by humans [1,2,3], and due to the abundance of seawater on Earth, it is expected to be an effective solution to the scarcity of freshwater resources through some emerging desalination technologies such as reverse osmosis (RO), thermal treatment (TP), electrodialysis (ED), and capacitive deionization (CDI) [4]. In contrast, capacitive deionization (CDI) has the advantages of low energy consumption and no pollution [5,6,7], so it is receiving more and more attention from researchers. The essence of CDI is based on the double electric layer (EDL) theory, where the chloride and sodium ions in the solution are adsorbed at the interface between the electrode and the electrolyte solution [8,9]. In addition, the electrodes can be regenerated by breaking the external voltage (or by shorting the electrodes), and finally, the ions adsorbed on the electrodes are released into the solution [10]. Currently, the most critical factor determining the performance of capacitive deionization is still the electrode active material, and it is important to develop electrode materials with excellent physical and chemical properties [11,12]. The current research on electrode materials mainly focuses on carbon-based materials, metal oxide materials, and conductive polymer materials. Metal oxide and conductive polymer materials store ions through the Faraday reaction, but there are some problems such as high electrode production cost, unstable circulation, and low conductivity. Carbon-based materials mainly include activated carbon [13,14], carbon nanotubes [15], carbon fibers [16], carbon aerogels [17], carbon flakes [18], and graphene [19]. These materials have a high specific surface area, stable chemistry, and ultra-high conductivity and are favored by engineers, among which carbon-based materials represented by activated carbon have been applied to commercialize capacitive deionization and provide human society with a constant source of clean fresh water [20,21].

Although the activated carbon electrode has excellent properties of carbon-based materials, it still has the problem that a large number of disordered micropores cannot immerse the solution, and the effective specific surface area is very low [22,23]. Recent studies have shown that layered porous materials with a large number of micropores guarantee a high specific surface area of the material. Since the mesopores provide channels for ion transfer and solution permeation, the mass transfer resistance is reduced. The micropores in the carbon-based material can be fully permeated with the help of the mesoporous channels and exhibit a larger adsorption capacity, and the lower mass transfer resistance makes it easy to obtain faster adsorption rates. Therefore, layered porous carbon with a good proportion of micropore and mesopore distribution can be used in capacitive deionization electrodes to obtain better performance [24,25]. Meanwhile, doping the carbon structure with heteroatoms is an effective way to increase the conductivity and improve the desalination performance [26]. Element N is usually used as the preferred dopant because of its abundant active centers and its ability to effectively modulate the electronic properties and surface chemistry of activated carbon electrodes [27,28,29]. 

Among the various methods of synthesizing layered porous carbon, the template method is considered to be the most common one. By relying on the template as the main body, it is possible to design the nanostructure, micromorphology, and pore structure of the material and realize the control of the pore size distribution and morphology of the prepared sample. Usually, the template method is divided into the hard template method and the soft template method. The skeleton structure is prepared using materials such as silica, metal oxides, zeolites, etc. as hard templates, which are mixed with carbon sources to prepare precursors, carbonized at high temperatures, and then the hard sphere templates are removed by erosion. Wei et al. [30] used a silica hard sphere template and phenolic resin as the carbon source to prepare porous carbon spheres with layered nanostructures in yolk shells at different temperatures and showed excellent capacitance and low transfer resistance in electrochemical tests. Zhang et al. [31] prepared core-shell hybridized zeolite imidazolate skeleton-derived layered carbon using zeolite as a hard template, 2-methylimidazole as a protective agent, and dopamine as a carbon source. After confined pyrolysis, anatase-like carbon (ANHC) with the ordered distribution of pores in micropores, large pore volume, and high nitrogen content (7.12%) was obtained, which promoted mass transfer and electrical conductivity and improved CDI performance. The soft template method generally uses block copolymers, polymer microspheres, and other materials. Using the synergistic interaction between the soft template and the carbon source, the sol-gel is assembled in the presence of a structural guide, after which the soft template dissolved and died out by itself after high temperature, leaving porous carbon with a layered structure. Zhang et al. [32] used PSBA as a template and induced the carbon source pyrrole under ice-water bath conditions with FeCl_3_·6H_2_O as an initiator polymerization on the PSBA surface to form PSBA/PPy core-shell structure precursors, and the precursors were carbonized at a high temperature to obtain mesoporous carbon nanospheres. However, the general hard template method requires the use of strongly corrosive hydrofluoric acid to treat the template, while the soft template method requires the involvement of reagents such as initiators and structure inducers. This makes the preparation process complicated and makes it difficult to achieve precise control of the pore structure. Therefore, there is an urgent need for a method that is simple to prepare, inexpensive, and can produce layered porous carbon with suitable medium and micropore size distribution on a large scale.

CaCO_3_ is a cheap and easily available industrial chemical. The temperature at which its pyrolysis occurs as a template is mainly concentrated at 600–800 °C. The use of nano CaCO_3_ as a hard template has been investigated in supercapacitor energy storage electrodes, lithium-ion battery electrodes, drug carriers, and antibiotic adsorption active materials. Shi et al. [33] synthesized 3D layered porous carbon with void space by dilute hydrochloric acid (HCl) assisted etching of self-assembled CaCO_3_ template method combined with citric acid as a carbon source by pyrolysis and applied it in the anode of lithium-ion battery. Yan et al. [34] successfully fabricated a new type of carbon bead with well-developed multilayer porosity by combining carbon thermal reduction and simple HCl treatment of CaCO_3_ encapsulated carbon beads. A large number of interconnected mesopores and macropores effectively promoted the diffusion of tetracycline from the solution to the surface and inside the carbon beads, possessing notable adsorption and removal effects. Guo et al. [35] prepared NPCS with pomegranate-like nanostructures by CaCO_3_ sphere template-induced self-activation using dopamine as the carbon precursor and applied them in a supercapacitor device by controlling the experimental conditions and thermogravimetric analysis. The template-induced self-activation mechanism was carefully investigated by combining mass spectrometry with controlled experimental conditions. Under optimized conditions, the prepared NPCS exhibited a large specific surface area (up to 1984 m^2^/g) and a high level of nitrogen doping (N, 7.57%). Ma et al. [36] prepared hydroxyapatite (HAP) particles with good dispersion and high surface area in an aqueous solution with the aid of a dipeptide hydrogel system using porous CaCO_3_ as a template. Porous hydroxyapatite particles have a large surface area, which facilitates the loading of model drugs. It also showed enhanced cell internalization efficiency and low cytotoxicity, which can be used as a slow-release drug carrier, demonstrating the stable adsorption performance of the porous carbon spheres synthesized by this method. The above study demonstrates that the use of CaCO_3_ nanostructured hard spheres templates is a convenient and effective way to prepare ordered layered porous carbon. It is reasonable to believe that its application in the preparation of active materials for capacitive deionization electrodes can also yield excellent performance.

In this study, three types of N-doped layered porous carbon prepared by high-temperature solid-state method at different pyrolysis temperatures were used as electrode materials. The structure, pore size distribution, and specific surface area of the prepared N-doped layered porous carbon are characterized. The electrochemical performance, the salt adsorption capacity during the actual desalination process and the desalination performance under different initial conditions are compared.

## 2. Experimental Descriptions

### 2.1. Formulation of N-Doped Layered Porous Carbon

Nano CaCO_3_ is chosen as the hard sphere template, C_12_H_22_O_11_ as the carbon source, and C_3_H_6_N_6_ as the nitrogen source, and the three are mixed for pyrolysis. C_12_H_22_O_11_ is dehydrated and condensed into caramel between 190 and 210 °C, which is uniformly wrapped on the surface of CaCO_3_ particles. CaCO_3_ pyrolysis produces CaO and CO_2_, and the remaining CaCO_3_ will be removed by adding HCl in subsequent experiments, so sucrose is the only source of carbon. The thermogravimetric analysis of CaCO_3_ is shown in Figure 1. The CaCO_3_ nanoparticles start to pyrolyze near 610 °C, and the main pyrolysis process is concentrated between 620 °C and 800 °C. At high temperatures, CaCO_3_ nanoparticles decompose into CO_2_ and CaO. On the one hand, the CaCO_3_ template decomposes to form a large number of hollow pores, and the carbon skeleton with large and medium pores is formed in this regard. On the other hand, the generated CO_2_ gas can further physically activate the corrosion of the carbon skeleton, etching a large number of microporous defects on porous carbon and increasing the specific surface area. C_3_H_6_N_6_ is commonly used to prepare nanosheet C_3_N_4_, which is also a very good source of nitrogen. By adding C_3_H_6_N_6_ to the raw material and introducing nitrogen atom doping, it can functionally modify the carbon material, increase the specific surface area of the active material, and improve the hydrophilicity of the material. The N-doped layered porous carbon prepared in this study using the CaCO_3_ hard template method has the following four advantages:(1)The hard template method is simple and inexpensive and can produce N-doped layered porous carbon in large batches. Moreover, the hard template does not need to be treated with a strong acid such as hydrofluoric acid after pyrolysis, and only dilute HCl can be used to remove the template residue, which can preserve a large number of nanostructures.(2)Previously, the CaCO_3_ template method has been studied in the preparation of active materials for waste gas adsorption and supercapacitor electrode materials, and it has been proven to be a feasible and excellent method.(3)The preparation of layered porous carbon with a graded structure and a suitable ratio of micropores to mesopores is a hot research direction in the structural design of carbon materials. The microporous structure prepared by the CaCO_3_ template method can help provide a large specific surface area for ion adsorption. The formed mesoporous structures serve as ion transport and transport channels, which can also enhance the accessibility of solutions to the micropores, reduce the mass transfer resistance of ions in solutions, and accelerate the ion adsorption rate.(4)Heteroatom doping is also one of the current research hotspots for the functional modification of carbon materials. Through the doping of N atoms, it can create microporous defects during the generation of the material, increase the specific surface area, and enhance the electrical conductivity and hydrophilicity of the material, all of which help to improve the ion adsorption properties of the material.
Figure 1TG curve of CaCO_3_ spheres.
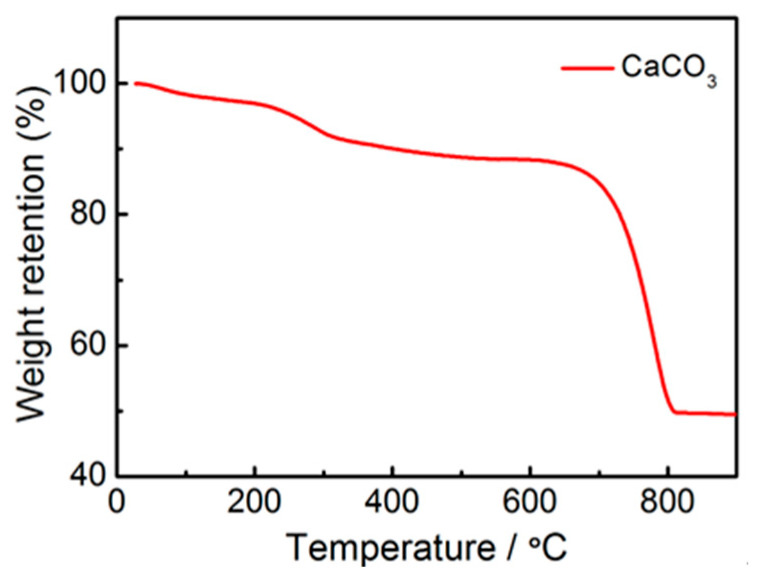


According to the results of the thermogravimetric analysis in Figure 1, the CaCO_3_ particles start to pyrolyze from 610 °C and the pyrolysis is finished by 800 °C. The activation temperature affects the degree of pyrolysis of the CaCO_3_ template, which in turn affects the CO_2_ physical activation of the carbon skeleton and pore channel generation. Therefore, the activation temperature is chosen as the main experimental variable to investigate the properties of N-doped layered porous carbon prepared by activation of CaCO_3_ templates at different temperatures. By pre-sintering experiments on CaCO_3_ particles between 600 °C and 900 °C with 50 °C intervals, the three temperatures of 650 °C at the beginning of pyrolysis, 750 °C at the halfway point of pyrolysis, and 850 °C after pyrolysis are chosen to investigate the optimal temperature for the preparation of N-doped layered porous carbon using CaCO_3_ as a template. 

YP-50F commercial super-activated carbon from Kuraray, Japan, is also used to prepare a desalination electrode for comparison in the CDI performance test phase. To evaluate the engineering potential of this CaCO_3_ self-activating template method for the preparation of N-doped hierarchical porous carbon.

### 2.2. Preparation of N-Doped Layered Porous Carbon

The raw materials required for the preparation of N-doped layered porous carbon are shown in Table 1.

The preparation process of N-doped layered porous carbon is as follows:(1)Commercial CaCO_3_ nanoparticles (40 nm), C_12_H_22_O_11_, and C_3_H_6_N_6_ are mixed in a mass ratio of 2:2:1 and ground for 1 h using an agate mortar to ensure that the raw materials are well mixed.(2)The mixed powder is fed into the powder compactor and pressed into cakes under 8 MPa pressure to reduce the voids between the raw materials. The small cakes of pyrolysis precursors are shown in Figure 2.(3)The pressed cakes are placed in a tube furnace with a zirconia crucible and heated to 650 °C, 750 °C, and 850 °C, respectively, at a heating rate of 5 °C/min in a high purity N_2_ atmosphere. After reaching the set temperature and then holding for 2 h, natural cooling down. The flow rate of nitrogen is set to 200 sccm.(4)The carbon product after sintering in the tube furnace is shown in Figure 3. The carbon product in the crucible is collected, put into an agate ball mill jar and placed in a planetary ball mill, and ground at 500 rpm for 4 h.(5)Collection of finer sintered products after grinding. Add excess 2 M HCl, stir magnetically for 4 h, and acid wash to remove CaO products and excess CaCO_3_.(6)The pickling solution is filtered using a polypropylene microporous filter membrane and a cloth funnel connected to a circulating water vacuum pump. Rinse repeatedly with anhydrous CH_3_CH_2_OH and deionized water alternately until the rinsed liquid is measured as neutral with PH test paper. The flushing is completed and the filter cake is collected.(7)The filter cake is placed in a vacuum drying oven and dried under vacuum at 85 °C for 12 h and collected to obtain N-doped layered porous carbon.
Figure 2Pyrolysis precursor piece.
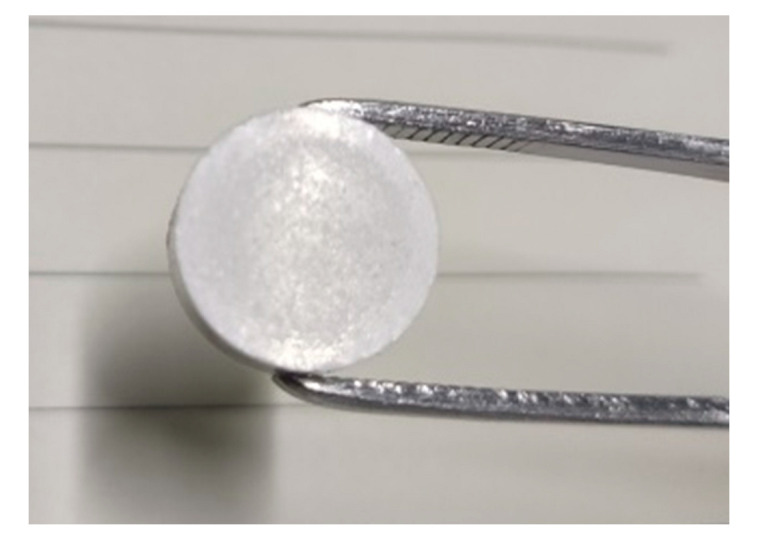

Figure 3Pyrolysis products.
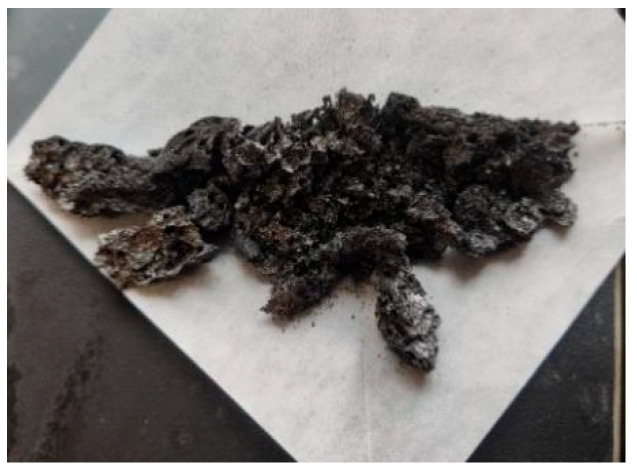


Three samples of N-doped layered porous carbon are obtained at 650 °C, 750 °C, and 850 °C, prepared and obtained according to the above process. The commercial super-activated carbon from Kuraray, Japan, is also named with its YP-50F product model as the experimental comparison sample.

### 2.3. Characterization and Electrochemical Measurements

The morphology of the N-doped layered porous carbon is examined by scanning electron microscopy (SEM, Regulus-8100, Hitachi Limited Co. Ltd., Tokyo, Japan). The porous characteristics are investigated by N_2_ adsorption-desorption isotherm. X-ray diffraction (XRD, XRD-6100, Shimadzu Research Laboratory (Shanghai) Co. Ltd., Shanghai, China) is used to analyze the structural information of the material. 

Electrochemical experiments are carried out using an electrochemical workstation (CHI608E, Shanghai Chenhua Instrument Co., Ltd, Shanghai, China). The area of the CV curve can be used to calculate the specific capacitance of the sample. The specific capacitance of the electrode is calculated by Equation (1), which is as follows:(1)$$C=\frac{\int IdU}{2\cdot \nu \cdot m\cdot U}$$
where *I* represents the instantaneous current value (A); *U* represents the voltage (V); *m* represents the mass of active material at the electrochemical electrode (g); *v* represents the scan rate (v/s); and Δ*U* represents the voltage window (V).

The performance of the synthesized materials was reported in terms of the salt adsorption capacity (SAC), which was calculated by Equation (2)
(2)$$Q=\frac{(C_{0}-C_{e})\cdot \nu_{NaCl}}{m}$$
where *C*_0_ and *C_e_* are the solution concentrations at initial and final equilibrium, respectively (mg/L); *Q* is the specific adsorption capacity of the CDI electrode (mg/g); *V_NaCl_* is the volume of the circulating solution during the CDI performance test (L) *m* is the mass of the active material in the CDI adsorption electrode (g).

## 3. Results and Discussion

### 3.1. Material Surface Morphology Analysis

The SEM results of samples at 650 °C, 750 °C, and 850 °C pyrolysis temperatures are shown in Figure 4, Figure 5 and Figure 6, respectively. 

The microscopic morphology of N-doped layered porous carbon is visible with the aid of scanning electron microscopy. At 5 μm resolution, it can be seen that the decomposition of porous carbon at 650 °C pyrolysis temperature has produced sporadic macropores formed on the carbon skeleton after the decomposition of the CaCO_3_ template can be observed at 1 μm resolution. At 5 μm resolution, the porous carbon at 750 °C pyrolysis temperature still has large stacks, and a considerable number of large holes could be observed at 1 μm resolution, indicating that CaCO_3_ continued to decompose as the temperature increased, creating more defects in the carbon skeleton. Layered porous carbon at 850 °C pyrolysis temperature, with almost complete disappearance of the bulk carbon skeleton at 5 μm resolution. Observing the picture at 1 μm resolution, the carbon skeleton is uniformly distributed with mesoporous structures left behind by CaCO_3_ self-activation, at which time all decomposition of the CaCO_3_ template is completed. As the pyrolysis temperature increases, the CaCO_3_ decomposes more, and more macropores and mesopores are formed. The continued decomposition of CaCO_3_ also produces more CO_2_ gas, which activates the etching of the porous carbon skeleton to create abundant defects and micropores, increasing the specific surface area of the material. The layered porous carbon at 850 °C pyrolysis temperature has the best-activated surface morphology and microscopic pore structure as seen in the SEM image.

Figure 7 shows the EDS spectra of the N-doped layered porous carbon at a pyrolysis temperature of 850 °C. It confirms that the experimental sample is composed of C, N, and O elements and no other elements are present. The doping content of N is 11.26% and the content of O is 7.55%. These oxygen doping concentrations may mainly come from the thermal decomposition of nano-CaCO_3_.

### 3.2. XPS Spectra of Materials

To verify the doping of N and O in the layered porous carbon, the materials are analyzed by XPS spectroscopy and the results are shown in Figure 8. It can be seen from Figure 8a that C, N, and O are present in all materials, and no other impurity elements are evident. Figure 8b–d show the XPS spectra of C1s, N1s, and O1s. The N and O elements are successfully doped. The C 1s, N 1s, and O 1s characteristic peaks are located around 285, 399, and 533 eV. It is further proved that N is successfully doped into carbon materials.

The XPS measurement results are shown in Table 2. From Table 2, the contents of C, N, and O in layered porous carbon at three different pyrolysis temperatures and the corresponding peak BE can be seen. The N content of the three samples is 13.41%, 12.21%, and 11.26%, respectively. Among them, the highest content of N-doped layered porous carbon N is found at 650 °C pyrolysis temperature.

### 3.3. Material Phase Analysis

The raw materials used to prepare the N-doped layered porous carbon are CaCO_3_, C_3_H_6_N_6_, and C_12_H_22_O_11_. The product is acid-washed and rinsed with large amounts of deionized water and anhydrous CH_3_CH_2_OH, and theoretically, the main element in the product is carbon. The physical phase of the N-doped layered porous carbon is analyzed using XRD, and the results confirmed that the main physical composition of the product was carbon. As shown in Figure 9, the prepared samples at 650 °C, 750 °C, and 850 °C all showed diffraction peaks at 2θ of 25 and 43.6, corresponding to the (002) and (100) crystallographic planes of graphitic carbon, respectively. It can be seen from the XRD plots that it has three typical powder patterns for turbostratic carbons. These materials lack the 3D periodicity of graphite and for this reason, there are two very broad unresolved peaks, and low intensity. The XRD diffraction atlas shows that the diffraction peak at 2θ = 25 is shifted to the left relative to that of crystalline graphite 2θ = 26.2, which shows that the graphite layer spacing increased. The preparation of layered porous carbon is doped with nitrogen. Impurity atoms increase the cell parameters and crystalline surface spacing. The width of the XRD peak is inversely proportional to the size of the grain. The smaller the diameter of the grain, the wider the corresponding XRD peak. Comparing the XRD of the samples at the three temperatures, it can be found that the diffraction peak at 850 °C has the weakest intensity, which indicates that it has a higher disordered carbon content. The higher the content of disordered carbon, the more defects in the layered porous carbon and the more active sites that provide ion adsorption.

### 3.4. Nitrogen Isothermal Adsorption and Desorption Determination

The specific surface area and pore structure of the three samples prepared in this paper are characterized by nitrogen isothermal adsorption and desorption determination. As shown in Figure 10a, all three samples have typical type IV isothermal adsorption curves. In the low-pressure region, the adsorption amounts of all three samples are rapidly increasing, indicating the presence of a large number of micropores in the material. The N-doped layered porous carbon at a pyrolysis temperature of 850 °C has a greater adsorption capacity and possesses more micropores than the other two prototypes. Between the relative pressure P/P_0_ = 0.45 and 0.9, the stage where mesopores play a role in adsorption and desorption, hysteresis loop lines appear in all three samples, indicating that the decomposition of CaCO_3_ creates abundant mesopores in the carbon skeleton and the CO_2_ produced has a good activation effect. The higher-pressure range (P/P_0_ > 0.9), is the stage where the adsorption and desorption of nitrogen by the macropores play a role. The three samples continue to show rapid growth in adsorption volume, which also corresponds to the structural morphology of the macropores photographed in SEM. The decomposition of CaCO_3_ leaves clear microporosity at the original location, and the maximum adsorption capacity of N-doped layered porous carbon at a pyrolysis temperature of 850 °C is also due to the complete decomposition of CaCO_3_, which produces the largest amount of microporosity, where sample 1 is the material without CaCO_3_ and carbonized at 850 °C. According to the isothermal adsorption and desorption curves, the specific surface area of the four samples is calculated using the BET (Brunaner-Emmett-Teller) model, and the N-doped porous carbon at a pyrolysis temperature of 850 °C possessed a specific surface area of 478 m^2^/g. In contrast, the N-doped porous carbon at 750 °C, and 650 °C pyrolysis temperatures are 370 m^2^/g and 355.9 m^2^/g. The CaCO_3_ template decomposes to form a large number of hollow pores, forming a carbon skeleton with large and medium pores. The generated CO_2_ gas can further corrode the carbon skeleton and etch a large number of microporous defects on the porous carbon, increasing the specific surface area. Therefore, the other three materials possess higher specific surface area compared to sample 1. The N_2_ isothermal adsorption and desorption curves show that the layered porous carbon micropores prepared at 850 °C, rich in mesopores and macropores, had the most developed pore structure. Figure 10b shows the pore size distribution curves of the three samples. It can be seen that 750 °C and 850 °C, in the 0–2 nm microporous fraction, possess a higher percentage, and the percentage of possession of the mesoporous fraction at 850 °C from 5 nm onwards consistently exceeds that of 750 °C and 650 °C. The results of N_2_ isothermal adsorption and desorption measurements show that higher temperatures allow sufficient pyrolysis of CaCO_3_ to create more macropores in the carbon skeleton and produce more CO_2_ gas to physically etch the carbon skeleton to form mesopores and micropores. The N-doped layered porous carbon prepared at 850 °C has the largest specific surface area and well-developed pore structure.

The pore distribution curves of BJH for the three materials are shown in Figure 10b. Most of the pores of the three materials are concentrated in the range of 2–70 nm. It is shown that all three materials possess good electrolyte accessibility. This is because the pore size of the electrode is greater than 2 nm, which facilitates the diffusion of electrolyte ions.

The pore structure characteristics of the samples at the three pyrolysis temperatures are shown in Table 3, including the BET-specific surface area, average pore width, and total pore volume, which are summarized in Table 3.

### 3.5. Electrochemical Performance Testing

Evaluation of the electrochemical properties of N-doped layered porous carbon prepared by this method using CHI608E electrochemical workstation. Cyclic voltammetric characteristics of the four electrode materials are tested with a three-electrode system.

Figure 11 shows the CV curves of 650 °C, 750 °C, 850 °C, and YP-50F electrode at a 5 mv/s sweep rate. As can be seen from the figure, the CV curves of all four samples show a rectangular-like shape, and no obvious redox peaks appeared. This indicates that the samples all exhibit pure capacitive properties rather than pseudocapacitive ones and that the materials are free of Faraday reactions in the electrochemical reactions. The electrode material at 850 °C pyrolysis condition has the largest CV curve area, 750 °C and YP-50F have almost the same CV curve area, and 650 °C has the smallest CV curve area.

At 850 °C, YP-50F, 750 °C and 650 °C, specific capacitances of the electrochemical electrodes are 83.8 F/g, 74.8 F/g, 70.2 F/g, and 42.7 F/g at a scan rate of 5 mv/s. At 850 °C, it has the largest specific capacitance under the same test conditions, indicating that the N-doped layered porous carbon material prepared at this temperature has the best ion storage capacity of the double layer.

The CV curves of the N-doped layered porous carbon at 850 °C pyrolysis temperature at 5, 10, 20, and 50 mv/s scan rates are shown in Figure 12. The CV curves all show a quasi-rectangular shape, indicating that the layered porous carbon has stable bilayer properties and good multiplicity performance. The specific capacitance of the N-doped layered porous carbon at 850 °C pyrolysis temperature is calculated according to Equation (1); 83.8 F/g, 76.9 F/g, 67.8 F/g, and 36.6 F/g at 5, 10, 20 and 50 mv/s, respectively. As the scanning rate increases, the specific capacitance of the same material decreases instead. This is because the faster the sweep rate is, the less time the ions have to diffuse along the pore channels to the internal pores of the electrodes, which in turn cannot be quickly transferred to the bilayer to form the capacitive layer.

### 3.6. CDI Desalination Experiment

In this section, commercial YP-50F super-activated carbon and three N-doped layered porous carbons with different pyrolysis temperatures are used as electrode materials. The procedure and data processing of typical CDI electrode desalination performance tests are described.

The YP-50F and three different N-doped layered porous carbon CDI electrodes with different pyrolysis temperatures are prepared. The mass of the YP-50F super activated carbon electrode is 0.2381 g and 0.2223 g, respectively, after weighing. The mass of three N-doped layered porous carbon electrodes with YP-50F error is not more than ±0.002 g. According to the formula used in the preparation of the electrodes, the mass percentage of the active material is 85%, so there is a total of 0.39134 g of active material in the two electrodes. The area of the two electrodes is 64.8 cm^2^, so the mass of active material in the YP-50F desalination electrode is about 6.13 mg/cm^2^.

The assembled CDI electrode assembly is rinsed with deionized water. The deionized water in the CDI electrode assembly is then shaken out. The peristaltic pump speed is set to 85 rpm (inlet solution flow rate of 15 mL/min), and the external 1.2 V DC voltage was connected. A conductivity meter is used to record the conductivity values during the absorption and desorption of the solution. The recorded conductivity values are shown in Figure 13a. The maximum and minimum concentrations of the solution throughout the adsorption process can be converted from the empirical conductivity-concentration Equation (2). The conductivity of the solution was 1012 μs/cm at the beginning with a concentration of 500 mg/L. When connected to 1.2 V dc, the conductivity of the solution started to decrease. Eventually, the conductivity of the solution plateaus. At this point, the concentration of the solution reaches a minimum and the adsorption of the CDI electrode is completed.

The salt adsorption capacities of YP-50F commercial activated carbon and N-doped layered porous carbon at three temperatures are calculated by Equation (2) as 9.66 mg/g, 12.56 mg/g, 8.89 mg/g, and 5.51 mg/g, respectively. As can be seen in Figure 13b, the best CDI desalination performance of 12.56 mg/g is obtained for the N-doped layered porous carbon at 850°C pyrolysis temperature under the same test conditions, which is better than the commercial super-activated carbon used as a comparison. The CDI desalination performance of N-doped layered porous carbon at 750 °C pyrolysis temperature is slightly lower than that of YP-50F commercial super-activated carbon with a salt adsorption capacity of 8.89 mg/g. The worst CDI performance of N-doped layered porous carbon at 650 °C pyrolysis temperature is 5.51 mg/g of salt adsorption capacity. The actual measured CDI desalination performance also corresponds to the previous material characterization and electrochemical results, further verifying the feasibility of the preparation scheme.

## 4. Results and Discussion

The influence of electrode material on the desalination of CDI modules is crucial. In addition, the applied voltage of the electrode, the inlet flow rate of the solution, and the initial solution concentration all affect the desalination performance of CDI modules. The sample with the best performance in N-doped layered porous carbon at 850 °C pyrolysis temperature is selected and compared with YP-50F to investigate the effect of changing operating parameters on the desalination performance of CDI.

### 4.1. Effect of External Voltage

When a capacitive deionization device is in operation, the applied voltage creates an electric potential on the surface of the desalination electrode, and the ions in solution are transported to the electrode particles and stored in the form of a double electric layer under the action of the electric potential field. The voltage, as the driving force for ion transfer, is a key influence on the performance of CDI desalination.

Prepare 60 mL of NaCl solution with the initial concentration of 500 mg/L, electrode plate spacing of 4 mm, and inlet water flow rate of 15 mL/min. The CDI performance curves of YP-50F and 850 °C N-doped layered porous carbon electrode are measured at room temperature (25 °C) under the applied 1.0 V, 1.2 V, and 1.4 V DC voltage. According to Equation (2), the specific adsorption desalination performance of YP-50F and 850 °C is calculated, and the results are shown in Figure 14.

It can be seen from Figure 14 that the specific adsorption amount of N-doped layered porous carbon and YP-50F increased as the applied voltage of the desalination electrode increased from 1.0 V to 1.2 V and then to 1.4 V. The specific adsorption amount and the applied voltage show a positive correlation. It is verified that the electric field intensity in the region of the two electrodes through which the solution flows increases with increasing operating voltage while the plate spacing of the CDI electrode is kept constant. The electrostatic adsorption is enhanced, and the electrode surface can store more ions and exhibit more excellent desalination performance. The specific adsorption capacity of YP-50F was 10.90 mg/g at the higher potential of 1.4 V, and that of 850 °C N-doped layered porous carbon reached 14.38 mg/g. The 850 °C N-doped layered porous carbon possesses better capacitive deionization performance than commercial YP-50F at three operating voltages.

### 4.2. Effect of Initial Solution Concentration

In the actual engineering application of CDI desalination electrodes, the concentration of the solution flowing into the electrode assembly varies with the nature of the liquid to be desalinated. In the case of desalinated seawater from CDI components, for example, the concentration of seawater in the same area can change due to ocean currents, tides, temperature, etc. The concentration of seawater also varies around the world, so this section explores the effect of initial solution concentration on the desalination performance of CDI.

The initial concentrations of 60 mL of NaCl solutions are 250 mg/L, 500 mg/L, and 1000 mg/L, respectively, with the electrode plate spacing of 4 mm and the inlet water flow rate of 15 mL/min. The CDI performance curves of YP-50F and 850 °C N-doped layered porous carbon electrode are measured at room temperature (25 °C) under the three concentrations of solutions with 1.2 V DC voltage applied. According to Equation (2), the specific adsorption desalination performance of YP-50F and 850 °C is calculated, and the results are shown in Figure 15.

As can be seen from Figure 15, the specific adsorption capacity of N-doped layered porous carbon and YP-50F increased as the initial solution concentration increased from 250 mg/L to 500 mg/L and then to 1000 mg/L. In the concentration range of 250 mg/L–1000 mg/L, the specific adsorption amount and the initial solution concentration show a positive correlation. On the one hand, the initial solution concentration increases and the number of ions contained per unit volume is higher, and it becomes easier for the electrode particles to capture and store ions. On the other hand, increasing the concentration of the solution leads to a solution with a higher dielectric constant and an increase in the thickness of the double layer, which in turn absorbs and stores more ions in it. Therefore, increasing the initial solution concentration appropriately can enhance the desalination performance of the electrode material. The salt adsorption capacities of YP-50F and N-doped layered porous carbon are 13.76 mg/g and 17.20 mg/g at an initial concentration of 1000 mg/L. The 850 °C N-doped layered porous carbon possesses better capacitive deionization performance than commercial YP-50F at three initial solution concentrations.

### 4.3. Effect of Solution Inlet Flow Rate

When the CDI desalination assembly is in operation, ions are transferred from the solution and eventually adsorbed into the electrical double layer. The transfer of ions is carried out by a combination of the macroscopic flow of the solution, its diffusion under the concentration gradient, and electric adsorption force traction in the potential field. By adjusting the speed of the circulating peristaltic pump, the inlet solution flow rate is changed to affect the macroscopic convective motion of ions in the solution. In this section, the effect of changing the inlet flow rate of the solution on the desalination performance of CDI is investigated.

Prepare 60 mL of NaCl solution with an initial concentration of 500 mg/L and electrode plate spacing of 4 mm. The CDI performance curves of the YP-50F and 850 °C nitrogen-doped layered porous carbon electrodes are measured at room temperature (25 °C) with an applied 1.2 V DC voltage at 8 mL/min, 15 mL/min and 25 mL/min inlet solution flow rates.

As can be seen from Figure 16, the salt adsorption capacity of the N-doped layered porous carbon and YP-50F decreased as the inlet solution flow rate increased from 8 mL/min to 15 mL/min and then to 25 mL/min. In the flow rate range of 8 mL/min–25 mL/min, the salt adsorption capacity shows a negative correlation with the inlet solution flow rate. The best salt adsorption effect is obtained for YP-50F and N-doped layered porous carbon at a low inlet solution flow rate of 8 mL/min. The salt adsorption capacity of YP-50F is 10.98 mg/g and that of N-doped layered porous carbon is 14.93 mg/g. CDI systems at low inlet flow rates can achieve better desalination. Because on the one hand, the lower the flow rate, the longer the ions flow over the electrode surface and are more easily captured by the electrode active material. On the other hand, with lower flow rates, the water flow disturbance effect is reduced, and the ions adsorbed on the electrodes are less likely to be hydraulically flushed back into the flow channel, exhibiting greater ion adsorption. At all three inlet solution flow rates, the N-doped layered porous carbon possesses better capacitive deionization performance than the commercial YP-50F.

In this paper, an N-doped layered porous carbon is successfully prepared using nano CaCO_3_ as a hard template, C_12_H_22_O_11_ as a carbon source, and C_3_H_6_N_6_ as a nitrogen source. The morphological and structural characteristics, electrochemical properties, and CDI properties under different pyrolysis conditions are systematically investigated. Compared with the commercial YP-50F super-activated carbon sample, the experimentally prepared N-doped layered porous carbon has more structural defects, higher specific capacitance, and better desalination capacity. The best performing 850 °C sample is selected for comparison with the commercial YP-50F supercapacitor activated carbon. The effects of three operating parameters, namely, applied electrode voltage, initial solution concentration, and inlet solution flow rate, on the desalination performance of CDI are studied. A larger applied voltage at the electrode, higher initial solution concentration, and lower inlet solution flow rate can enhance the performance of CDI desalination for both materials. In the experiments exploring the effect of operating parameters on the desalination performance, the N-doped layered porous carbon at 850 °C pyrolysis temperature consistently outperformed the CDI desalination performance of YP-50F. Therefore, the method proposed in this paper is a scheme for the preparation of CDI electrode active materials with reliable pore structure design. At the same time, with the cheap and easily available raw materials, simple preparation process, and excellent CDI desalination performance of the products, this method is a promising solution for the commercialization of material preparation.

## Figures and Tables

**Figure 4 materials-16-01435-f004:**
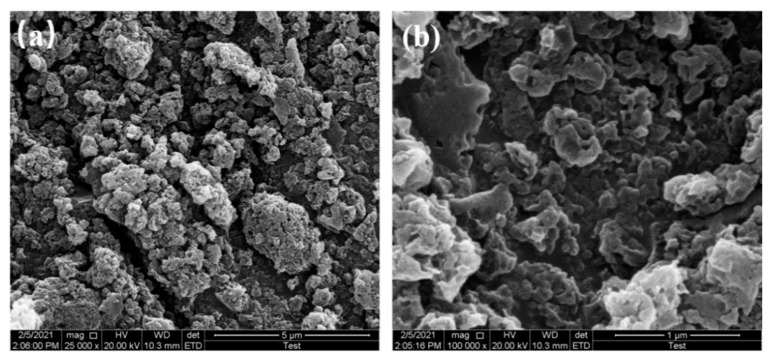
SEM images of 650 °C at (**a**) 5 μm resolution, (**b**) 1 μm resolution.

**Figure 5 materials-16-01435-f005:**
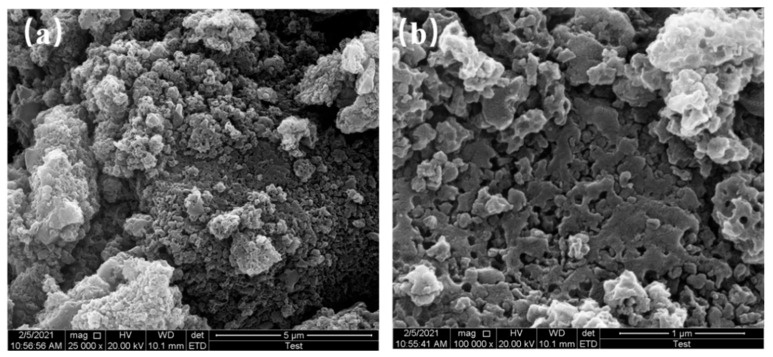
SEM images of 750 °C at (**a**) 5 μm resolution, (**b**) 1 μm resolution.

**Figure 6 materials-16-01435-f006:**
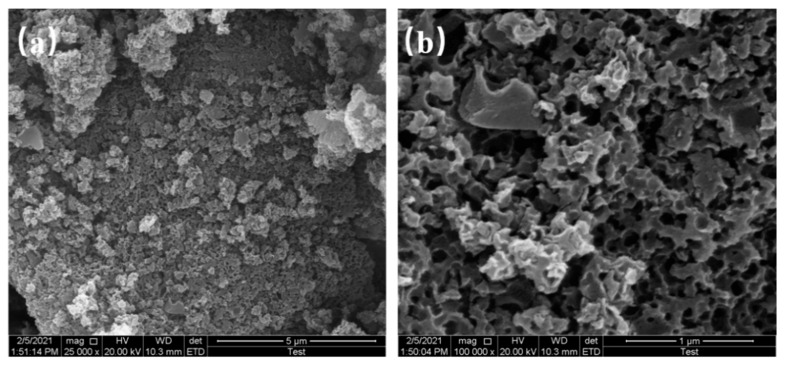
SEM images of 850 °C at (**a**) 5 μm resolution, (**b**) 1 μm resolution.

**Figure 7 materials-16-01435-f007:**
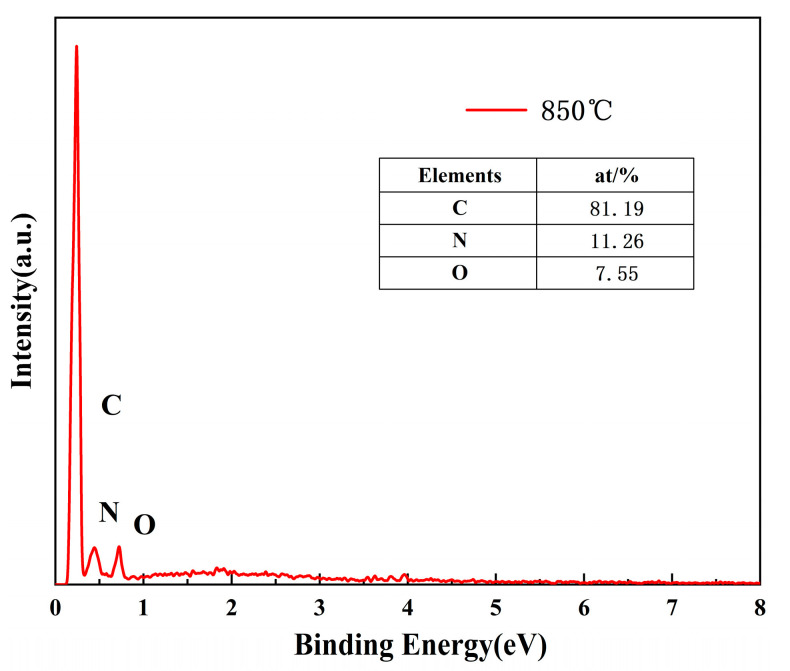
EDS spectrum of N-doped layered porous carbon at 850 °C.

**Figure 8 materials-16-01435-f008:**
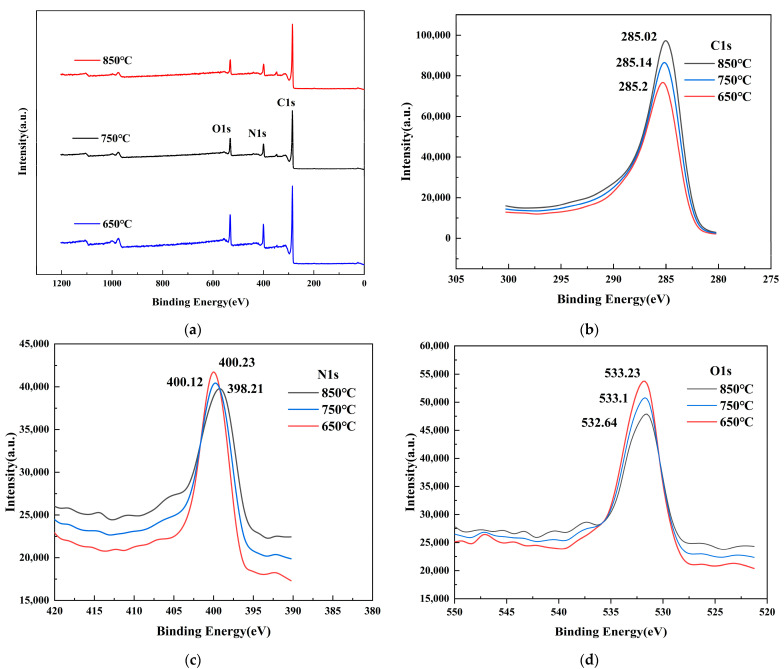
(**a**) XPS spectra of materials; XPS spectra of (**b**) C 1 s, (**c**) N 1 s, and (**d**) O 1 s.

**Figure 9 materials-16-01435-f009:**
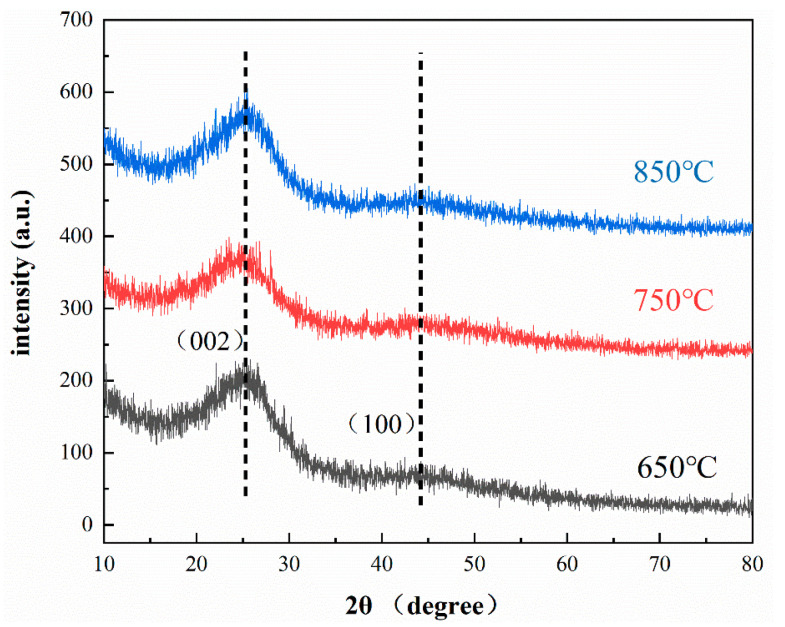
XRD spectrum of N-doped layered porous carbon at 650 °C, 750 °C and 850 °C.

**Figure 10 materials-16-01435-f010:**
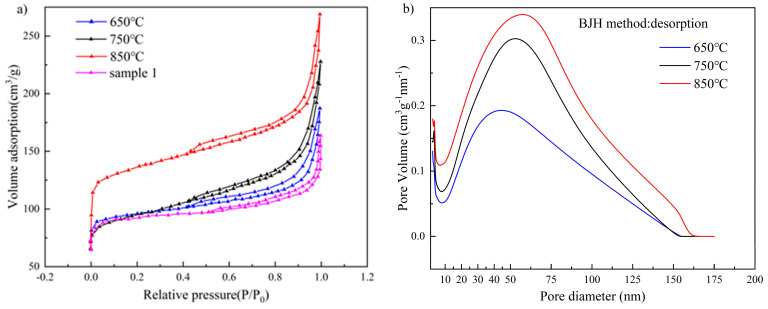
(**a**) N_2_ adsorption-desorption isotherms curves and (**b**) pore size distribution at 650 °C, 750 °C, and 850 °C.

**Figure 11 materials-16-01435-f011:**
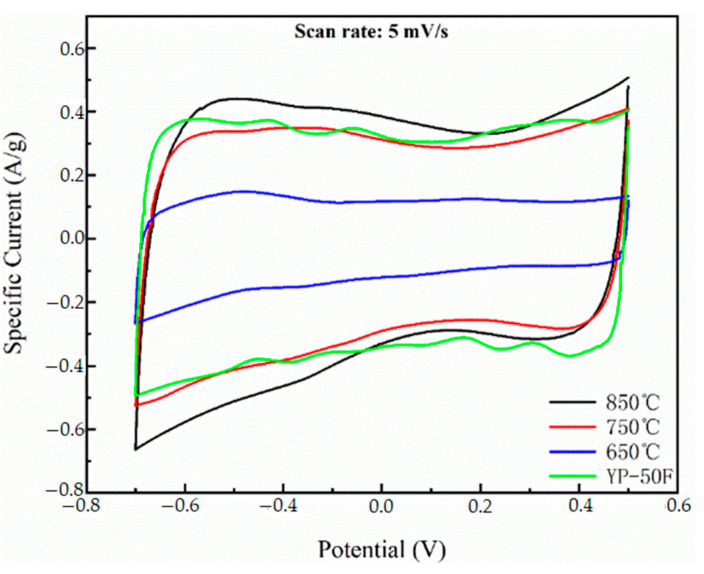
CV curves of 650, 750, 850, and YP-50F at a scanning rate of 5 mV/s.

**Figure 12 materials-16-01435-f012:**
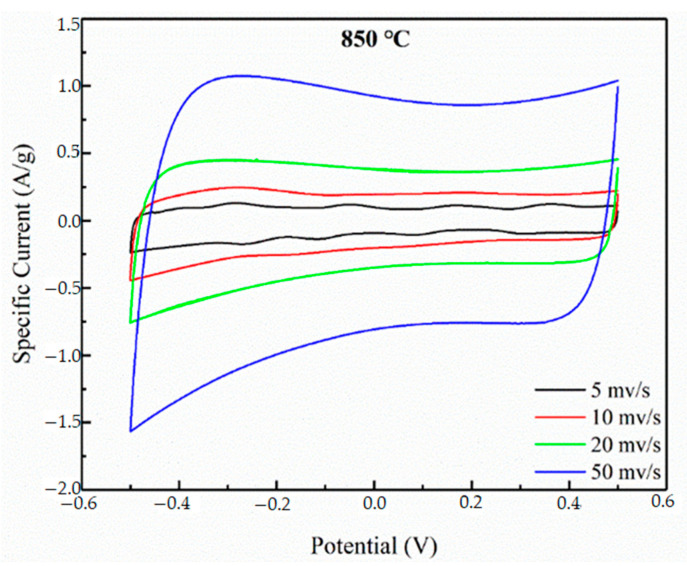
CV curves at 850 at scanning rates of 5 mV/s, 10 mV/s, 20 mV/s, and 50 mV/s.

**Figure 13 materials-16-01435-f013:**
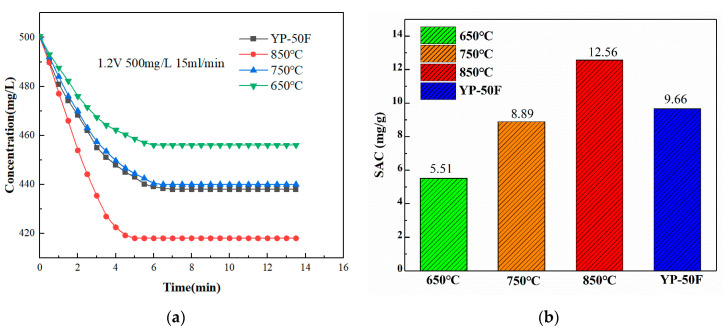
(**a**) Concentration-time curves of YP-50F and N-doped layered porous carbon. (**b**) SAC of YP-50F and N-doped layered porous carbon.

**Figure 14 materials-16-01435-f014:**
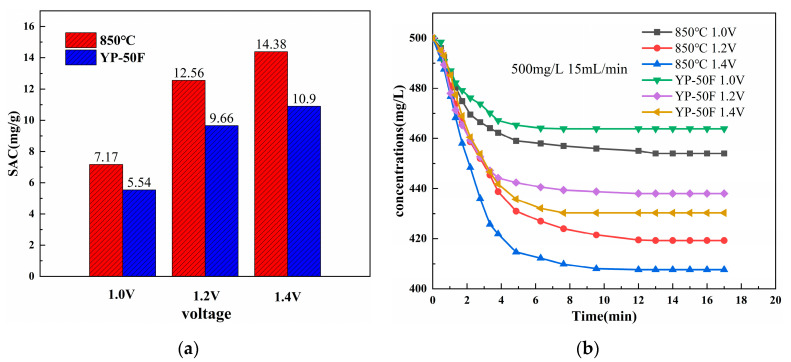
(**a**) SAC of YP-50F and N-doped layered porous carbon at 850 °C pyrolysis temperature. (**b**) Concentration-time curves of N-doped layered porous carbon at YP-50F and 850 °C pyrolysis temperature under different voltage conditions.

**Figure 15 materials-16-01435-f015:**
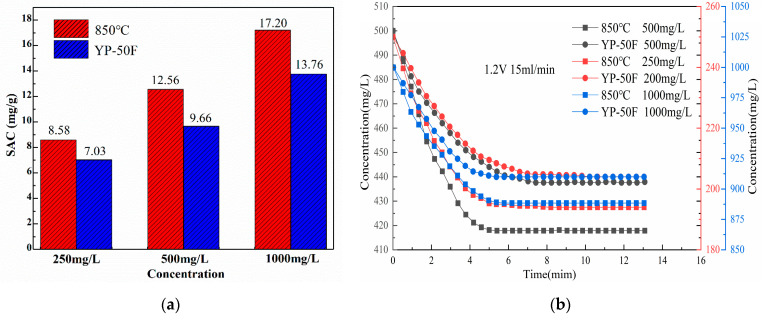
(**a**) SAC of YP-50F and N-doped layered porous carbon at 850 °C pyrolysis temperature. (**b**) Concentration-time curves of N-doped layered porous carbon at YP-50F and 850 °C pyrolysis temperature under different concentration conditions.

**Figure 16 materials-16-01435-f016:**
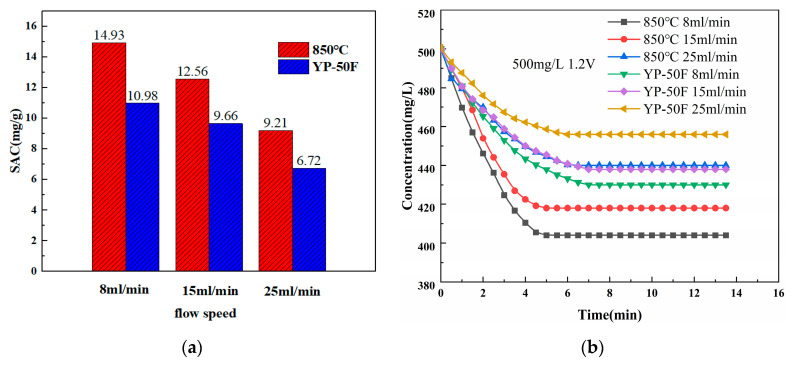
(**a**) SAC of YP-50F and N-doped layered porous carbon at 850 °C pyrolysis temperature. (**b**) Concentration-time curves of N-doped layered porous carbon at YP-50F and 850 °C pyrolysis temperature under different flow rate conditions.

**Table 1 materials-16-01435-t001:** Experimental reagents and materials.

Reagent/Material Name	Purity/Specification	Manufacturers
NaCl	AR	Sinopharm Chemical Reagent Co., Ltd. (Suzhou, Jiangsu, China)
Calcium carbonate (CaCO_3_)	40 nm	Shanxi Xintai Nano Material Company (Yuncheng, Shanxi, China)
Sucrose (C_12_H_22_O_11_)	AR	Sinopharm Chemical Reagent Co., Ltd. (Suzhou, Jiangsu, China)
Melamine (C_3_H_6_N_6_)	AR	Shanghai Aladdin Biochemical Technology Co., Ltd. (Shanghai, China)
Hydrochloric acid (HCl)	AR	Huadong Medicine Co., Ltd. (Zhejiang, China)
Super activated carbon	YP-50F	Japan Kuraray Co., Ltd. (Shanghai, China)
Polytetrafluoroethylene (PTFE)	D-210C	Daikin Fluorochemicals (China) Co., Ltd. (Shanghai, China)
Ethanol (CH_3_CH_2_OH)	AR	Sinopharm Chemical Reagent Co., Ltd. (Suzhou, Jiangsu, China)
Nitrogen(N_2_)	99.99%	Danyang Shenghe Industrial Gas (Danyang, Zhenjiang, Jiangsu)
Polypropylene microporous filter membrane	70 mm	Haiyan New Oriental Plastics Technology Co., Ltd. (Ningbo, Zhejiang, China)

**Table 2 materials-16-01435-t002:** XPS measurement results.

Sample	From Element Analysis			Peak BE		
	at%C	at%N	at%O	C1s	N1s	O1s
650 °C	75.25	13.41	11.34	285.2	400.23	533.23
750 °C	78.35	12.21	8.44	285.14	400.12	533.1
850 °C	81.19	11.26	7.55	285.02	398.21	532.64

**Table 3 materials-16-01435-t003:** Values of surface area (S_BET_), total pore volume (V_BJH_), pore diameter (D_BJH_), and micropore volume (V_M_) of three samples.

Sample	S_BET_ (m^2^/g)	D_BJH_ (nm)	V_BJH_ (cm^3^/g)	V_M_ (cm^3^/g)
650 °C	478	4.721	0.282	0.095
750 °C	355.9	4.796	0.312	0.056
850 °C	325.6	5.864	0.333	0.036

## Data Availability

Not applicable.

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
