# Peer review of "Preparation of N-Doped Layered Porous Carbon and Its Capacitive Deionization Performance"

_materials, 2023, doi:10.3390/ma16041435_

Round 1

Reviewer 1 Report

The article deals with the actual topic of obtaining porous carbon materials. The authors have carried out significant experimental work and determined the main parameters of the obtained materials.

 However, there are several comments to the text of the paper:

1.            How do the authors prove that calcium ions (calcium compounds) are removed from the surface of the target material?

2.            How was the incorporation of nitrogen into the structure of the composites confirmed? Were the materials examined by IR and Raman spectroscopy?

3.            Line 369 is a duplication of the Results and discussion section. Perhaps another section corresponding to further subheadings should be indicated.

4.            The References should be brought to the requirements of the publisher.

 After making the necessary changes the article could be published.

Author Response

Thank you for your review comments, which we have responded to in the attached document.

Reviewer 2 Report

Authors have described the synthesis of N-doped carbon based porous structure from natural sources and its implementation in deionization process. The manuscript is well written and have practical applications like water desalination.

Therefore I recommend the manuscript for publication. But before final acceptance, authors should answer the following comments.

(1) Figure 1 tells about the thermal decomposition of nano CaCO3 where at 800 °C, the material have been degraded more than 50%. But authors are using 850 °C for making best performing N doped porous carbon structures. I hope there is no other carbon source other than sucrose is interfering.

(2) Can author make an elemental analysis of the material to make sure there are no metal atoms/ ions presence in the sample. An EDAX analysis of the microstructure will be sufficient to prove the statement.

(3)The reference format of the manuscript is pretty bad. There are no journal names in the reference. This is very unprofessional. Authors should be very careful regarding these mistakes.

Author Response

(The authors gave the same response as above.)

Reviewer 3 Report

This important paper should be brought to higher level of English.

The dots are missing in several places in the text.

Chemical formulas should be written with indexes (for example, CaCO3).

I suggest to spread the sentence: Carbon-based materials mainly include activated carbon [13, 14], carbon nanotubes[15, DOI:10.3390/ma14061428], carbon fibers[16], carbon aero-gels[17], carbon flakes [DOI:10.3390/polym12122766] and graphene[18].

Please, mention the producers of the material sources.

Please, improve the graph at the figure 14b.

Please, correct: „Figure.1 TG curves…“ – there is only 1 curve.

Author Response

(The authors gave the same response as above.)
